

# Heat shock protein 90 is involved in the regulation of HMGA2-driven growth and epithelial-to-mesenchymal transition of colorectal cancer cells

Chun-Yu Kao[1], Pei-Ming Yang[2],*, Ming-Heng Wu[3],*, Chi-Chen Huang[4],* Yi-Chao Lee[4] and Kuen-Haur Lee[2]

[1] Department of Pediatric Surgery, Taipei Medical University-Shuang Ho Hospital, New Taipei City, Taiwan
[2] Graduate Institute of Cancer Biology and Drug Discovery, College of Medical Science and Technology, Taipei Medical University, Taipei, Taiwan
[3] Graduate Institute of Translational Medicine, College of Medical Science and Technology, Taipei Medical University, Taipei, Taiwan
[4] Graduate Institute of Neural Regenerative Medicine, College of Medical Science and Technology, Taipei Medical University, Taipei, Taiwan
* These authors contributed equally to this work.

Corresponding authors
Yi-Chao Lee, yclee@tmu.edu.tw
Kuen-Haur Lee, khlee@tmu.edu.tw

## ABSTRACT

High Mobility Group AT-hook 2 (HMGA2) is a nonhistone chromatin-binding protein which acts as a transcriptional regulating factor involved in gene transcription. In particular, overexpression of HMGA2 has been demonstrated to associate with neoplastic transformation and tumor progression in Colorectal Cancer (CRC). Thus, HMGA2 is a potential therapeutic target in cancer therapy. Heat Shock Protein 90 (Hsp90) is a chaperone protein required for the stability and function for a number of proteins that promote the growth, mobility, and survival of cancer cells. Moreover, it has shown strong positive connections were observed between Hsp90 inhibitors and CRC, which indicated their potential for use in CRC treatment by using combination of data mining and experimental designs. However, little is known about the effect of Hsp90 inhibition on HMGA2 protein expression in CRC. In this study, we tested the hypothesis that Hsp90 may regulate HMGA2 expression and investigated the relationship between Hsp90 and HMGA2 signaling. The use of the second-generation Hsp90 inhibitor, NVP-AUY922, considerably knocked down HMGA2 expression, and the effects of Hsp90 and HMGA2 knockdown were similar. In addition, Hsp90 knockdown abrogates colocalization of Hsp90 and HMGA2 in CRC cells. Moreover, the suppression of HMGA2 protein expression in response to NVP-AUY922 treatment resulted in ubiquitination and subsequent proteasome-dependant degradation of HMGA2. Furthermore, RNAi-mediated silencing of HMGA2 reduced the survival of CRC cells and increased the sensitivity of these cells to chemotherapy. Finally, we found that the NVP-AUY922-dependent mitigation of HMGA2 signaling occurred also through indirect reactivation of the tumor suppressor microRNA (miRNA), let-7a, or the inhibition of ERK-regulated HMGA2 involved in regulating the growth of CRC cells. Collectively, our studies identify the crucial role for the Hsp90-HMGA2 interaction in maintaining CRC cell survival and migration. These findings have

significant implications for inhibition HMGA2-dependent tumorigenesis by clinically available Hsp90 inhibitors.

## INTRODUCTION

High Mobility Group AT-hook (HMGA) nonhistone chromatin-binding proteins, including HMGA1 (isoforms HMGA1a and HMGA1b) and HMGA2, are architectural nuclear factors involved in chromatin remodeling and gene transcription (*Reeves & Nissen, 1990*). HMGA1 and HMGA2 have similar functions and are abundantly expressed in the early embryo, in which cells proliferate rapidly (*Sgarra et al., 2004*). However, *HMGA2* cannot be detected in adult human tissues, in which it is probably completely silenced (*Gattas et al., 1999*; *Rogalla et al., 1996*). In particular, HMGA2 is weakly expressed only in preadipocytic proliferating cells (*Anand & Chada, 2000*) and spermatocytes (*Di Agostino et al., 2004*). Conversely, several studies have reported that the association of HMGA2 overexpression with the transformation and metastatic progression of neoplastic cells suggests its causal role in carcinogenesis and tumor progression (*Mahajan et al., 2010*; *Piscuoglio et al., 2012*; *Wang et al., 2011*; *Wend et al., 2013*; *Xu et al., 2004*). Furthermore, the essential role of HMGA2 in cell proliferation and migration has been reported in various cancers (*Malek et al., 2008*; *Sun et al., 2013*; *Xia et al., 2015*; *Yang et al., 2011*). Thus, the HMGA2 protein is a promising biomarker for cancer detection as well as a potential molecular target in cancer therapy.

Heat shock protein 90 (Hsp90), one of the most abundant and highly conserved molecular chaperones, is essential for the stability and function of multimutated, chimeric, and overexpressed signaling proteins that promote the growth, mobility, and survival of cancer cells (*Neckers, 2002*). In addition, Hsp90 is involved in the maturation and stabilization of various oncogenic client proteins crucial for oncogenesis and malignant progression (*Chiosis, Caldas Lopes & Solit, 2006*). Thus, Hsp90 is considered a valuable target for cancer therapy. Moreover, using a combination of microarray gene expression of 132 Colorectal Cancer (CRC) patients and Connectivity Map data mining, extremely strong positive connections were observed between Hsp90 inhibitors and CRC, which indicated their potential for use in CRC treatment (*Su et al., 2015*). However, the correlation and regulatory mechanism between Hsp90 and HMGA2 in CRC remain largely unclear.

## MATERIALS AND METHODS

### Chemicals, reagents, antibodies, and expression constructs

NVP-AUY922 was purchased from Selleck Chemicals LLC (Houston, TX, USA). Crystal violet and DMSO were obtained from Sigma (St. Louis, MO, USA). Small interfering RNA (siRNA) targeting Hsp90 or HMGA2 mRNA, control siRNA, and the RNAiMax transfection reagent were purchased from Life Technologies (Carlsbad, CA, USA). Rabbit

antibodies against Hsp90, CDK4, E-cadherin, vimentin, Twist, Snail, Slug, extracellular signal-regulated kinase (ERK), Thr(P)202/Tyr(P)204-ERK1/2, cAMP response element-binding protein (CREB), Ser(P)133-CREB, focal adhesion kinase (FAK), Tyr(P)397-FAK, Lin28B, Tyr(P)705-Stat3, Stat3, and c-Myc were obtained from Cell Signaling (Beverly, MA, USA). HMGA2 and GFP were obtained from Santa Cruz Biotechnology (Santa Cruz, CA, USA). Mouse monoclonal antibody against $\beta$-actin was purchased from MP Biomedicals (Irvine, CA, USA). AZD6244 was acquired from Selleckchem (Houston, TX, USA).

## Cell culture

CRC cell lines were provided by Prof. YW Cheng and Prof. H Lee (Graduate Institute of Cancer Biology and Drug Discovery, Taipei Medical University). Stable DLD-HMGA2-GFP expression cell line was provided by Dr. PM Yang (Graduate Institute of Cancer Biology and Drug Discovery, Taipei Medical University). All CRC cell lines were cultured in RPMI-1640 and supplemented with 10% Fetal Bovine Serum (FBS) and 1% Penicillin and Streptomycin (P/S). CRL-1459/CCD-18Co (noncancerous human colon cells) was provided by Prof. PJ Lu (Institute of Clinical Medicine, National Cheng Kung University) and cultured in minimum essential Eagle's medium and supplemented with 10% FBS and 1% Penicillin and Streptomycin (P/S).

## Cell viability assay

Cell viability was determined through crystal violet staining, as described by (*Kim, Talanian & Billiar, 1997*). In brief, the cells were plated in 96-well plates at 4000 cells/mL and subjected to DMSO or NVP-AUY922 treatment at the indicated concentrations. Viable cells were stained with 0.5% crystal violet in 30% ethanol for 10 min at room temperature. Subsequently, the plates were washed four times with tap water. After drying, the cells were lysed with a 0.1 M sodium citrate solution, and the dye uptake was measured at 550 nM using a 96-well plate reader. Cell viability was calculated by comparing the relative dye intensities of the treated and untreated samples.

## Tissue microarray of CRC clinical specimens

A colon adenocarcinoma tissue array was purchased from US Biomax (CO1505, containing 50 cases of CRC tissues with matched adjacent tissues as the controls). All tissue sections were stained using a standard Immunohistochemical (IHC) protocol. In brief, slides were deparaffinized using serial xylene–ethanol treatment. Antigens were retrieved through boiling in a sodium citrate buffer for 10 min. Slides were blocked in 5% normal goat serum for 1 hour at room temperature. After blocking, the slides were incubated with a primary antibody against HMGA2, followed by a biotin-conjugated secondary antibody, Horseradish Peroxidase polymer (HRP), and a diaminobenzidine-tetrahydrochloride-dihydrate solution. The staining intensity was scored as follows: 0 point, negative; 1 point, weakly positive; 2 points, moderately positive; and 3 points, strongly positive.

## Quantitative reverse-transcription polymerase chain reaction

Total RNA was extracted from the cell lines with or without drug treatment using a Qiagen RNeasy kit and Qiashredder columns according to manufacturer instructions

(Valencia, CA, USA). One microgram of the total RNA was reverse transcribed to cDNA using a SABiosciences Reaction Ready™ First Strand cDNA Synthesis Kit (Frederick, MD, USA). Quantitative Reverse-Transcription Polymerase Chain Reaction (RT-PCR) was performed in an Applied Biosystems StepOne Plus™ Real-Time PCR System (Foster City, CA, USA) using an automated baseline and threshold cycle detection. For detecting the expression levels of let-7a, HMGA2, and GAPDH, the amplification and detection of specific products were performed using the cycle profile of the Qiagen miScript SYBR green PCR starter kit (Valencia, CA, USA). The relative gene expression level was calculated by comparing the cycle times for each target PCR. The let-7a PCR Ct values were normalized by subtracting the U6 rRNA Ct value (internal control) (RNU6-2_11 miScript Primer Assay; Qiagen catalog number: MS00033740). The sequences of the primers used in this study are listed as follows: Let-7a: 5′-UGAGGUAGU AGGUUGUAUAGUU-3′; HMGA2: forward 5′-TTCAGCCCAG-GGACAACCT-3′ and reverse 3′-TCTTGTTTTTGCTGCCTTTGG-5′; GAPDH: forward 5′-AATCCCATCACCA TCTTCCA-3′ and reverse 3′-ACTCATGCAGCACCTCAGGT-5′.

## Transfection

Cells were transfected with siRNAs for 48 hours using Lipofectamine 2000 (Life Technologies) according to the manufacturers' instructions. The siRNA used in this study from Life Technologies and their sequences were as follows: Hsp90 (siRNA ID: s6994): sense 5′-CUAUGGGUCGGUGGAACAAAtt-3′ and antisense 5′-UUUGUUCCACGA CCC- AUAGgt-3′; HMGA2 (siRNA ID: s224869): sense 5′-GGAGAAAAACGGCAAG AGtt-3′ and antisense 5′-CUCUUGGCCGUUUUUCUCCag-3′.

## Immunofluorescence staining

HCT116 cells grown on glass coverslips were transfected with control siRNA or siHsp90, respectively. At 48 hours post-transfection, the cells were fixed with 4% ice-cold paraformaldehyde at 4 °C for 20 min and then permeabilized with PBS with 0.5% Triton X-100 for 10 min at room temperature (RT), then washed, and blocked with 10% goat serum in Phosphate-Buffered Saline (PBS) for 45 minutes at RT. Cells were then incubated overnight at 4 °C with the mouse anti-Hsp90 (1:300; Abcam, MA, USA) or rabbit anti-HMGA2 (1:300; Santa Cruz Biotechnology, TX, USA). After washing, the cells were incubated at RT for 1.5 hours with Alexa-Fluor-546-conjugated goat anti-mouse IgG secondary antibody (1:500) (Invitrogen, Carlsbad, CA, USA) or Alexa-Fluor-488-conjugated goat anti-rabbit IgG secondary antibody (1:500) (Invitrogen, Carlsbad, CA, USA). After 3 washes, cells were mounted on glass slides in Mount medium containing DAPI (4, 6 diamidino-2-phenylindole;Polysciences) (Vector Laboratories, CA, USA). The images were examined on an Olympus FV1000 confocal microscope (Olympus Corp., Tokyo, Japan).

## Immunoprecipitation

The interaction between Hsp90 protein and HMGA2 was studied by immunoprecipitation analysis of extracts prepared from HCT116 or DLD1 cell lines. Cells were lysed, incubated in IP lysis buffer (10 mM Tris-HCl pH 7.5, 150 mM NaCl, protease inhibitor

mixture) for 30 min on ice, and then sonicated (3 times for 10 sec each). After centrifugation at 1400 g for 5 min at 4 °C, the supernatants were collected from each sample and then pre-cleared by incubation with 50% protein A/G agarose beads in the IP lysis buffer at 4 °C for 1 hour with rocking. After removal of the protein A/G beads by centrifugation, protein concentration in each sample was measured and aliquots containing 500 μg of protein were incubated with primary antibodies overnight at 4 °C. The immunoprecipitates bound to the protein A/G–Sepharose beads were washed, boiled and analyzed by immunoblotting.

## Western blotting

Cell lines were placed in a lysis buffer at 4 °C for 1 hour. Protein samples were electrophoresed using 8%–15% SDS-polyacrylamide gel electrophoresis, as described by *Su et al. (2015)*.

## Human phospho-kinase array

HCT116 cell line was analyzed in the array panel of kinase phosphorylation profiles after DMSO or NVP-AUY922 treatment (Human Phospho-Kinase Array, ARY003; R&D Systems). This array specifically screens for relative phosphorylation levels of 42 individual proteins involved in cellular proliferation and survival. Each phospho-kinase array has duplicate signal spots for each gene. After DMSO or NVP-AUY922 (10 nM) treatment, cell lysates were incubated with the membrane. Thereafter, a cocktail of biotinylated detection antibodies, streptavidin-HRP, and chemiluminescent detection reagents were used for detecting phosphorylated proteins. The bar graphs were normalized by using blank spot. The dot density was scanned from the scanned X-ray film, and images were analyzed and quantified using image analysis software (NIH-Image J).

## In vitro migration assay

Assays were performed using Falcon™ cell culture inserts (8-μm pore size) in a 24-well plate (BD Biosciences, San Jose, CA, USA) according to manufacturer instructions. In the migration assay, HCT116 cells ($10^4$ cells/well) in 0.5 mL of serum-free medium were seeded onto the upper chamber membranes that received different treatment. These membranes were previously inserted into the 24-well plates containing 10% FBS-supplemented medium. After 24 hours, the cells were fixed with 100% methanol and stained with 5% Giemsa stain (Merck, Darmstadt, Germany). Nonmigrated cells that remained in the upper chambers were removed by wiping the top of the inserted membranes using a damp cotton swab, leaving only those cells that migrated to the underside of the membranes. All experiments were performed in triplicate and photographed under a phase-contrast microscope (200×).

## Statistical analysis

Statistical analyses were performed as recommended by an independent statistician. Unpaired Student's *t*-test was used for analyses. All statistical analyses were performed using SPSS (SPSS, Chicago, IL, USA), and all values are expressed as the mean ± standard deviation. $p < 0.05$ was considered statistically significant.

## RESULTS

### Elevated expression of HMGA2 mRNA and protein in CRC cell lines and tissues

To determine the HMGA2 expression levels in these CRC patients, we first analyzed the gene expression of *HMGA2* in 132 CRC tumor samples: 67 primary CRC tissues, 65 metastatic tissues, and nine normal colon controls. As expected, HMGA2 expression was significantly upregulated in metastatic and primary CRC tissues compared with that in the normal colon controls (Fig. S1A). Similarly, the Hsp90 expression levels were analyzed in the same metastatic and primary CRC tissues, and the mRNA expression levels of Hsp90 were similar to those of HMGA2 (Fig. S1B). Next, to further understand the level of HMGA2 gene expression in human cancers, various cancer cell lines were selected from the National Cancer Institute Cancer Genome Anatomy Project gene expression database. CRC cell lines had relatively high levels of HMGA2 mRNA expression (Fig. 1A). Notably, HMGA2 was highly expressed in CRC cell lines among the 9 different types of cancers, thus validating its specificity in CRC. To determine the level of HMGA2 mRNA and protein expression in CRC cell lines, eight CRC cell lines and one noncancerous human colon cell line (CRL-1459) were chosen. Compared with CRL-1459, high expression of HMGA2 mRNAs and proteins were in all CRC cell lines except SW480 (Figs. S2 and 1B). Thus, the high levels of HMGA2 protein expression in HCT116 cells was attributed to its high mRNA expression. Therefore, HCT116 was chosen for further cell model experiments. HMGA2 protein expression was further examined in a colon adenocarcinoma tissue array (BioMax, Rockville, MD, USA). As shown in Fig. 1C, HMGA2 was upregulated in colon adenocarcinoma tissues of different tumor grades. The staining intensity of HMGA2 expression levels were defined on the basis of immunoexpression, as outlined in the IHC protocol (Fig. S3), and the colon adenocarcinoma tissues of all grades exhibited positive staining for HMGA2 (Fig. 1D). These results indicate that HMGA2 expression was specific and elevated in CRC cells.

### Effects of gene-specific inhibition of HMGA2 or Hsp90 and pharmaceutical inhibition of Hsp90 were similar

Specific knockdown of HMGA2 inhibited cell proliferation, leading to an epithelial-mesenchymal transition in human pancreatic cancer cells (*Watanabe et al., 2009*). To determine the effects of HMGA2 inhibition in CRC cells, cell proliferation and cell migration assays were performed; siHMGA2 knockdown significantly reduced mRNA expression level of HMGA2 (Fig. S4A) and the proliferation rate of siHMGA2-transfected HCT116 cells (Fig. 2A). The migration transwell assay was performed to determine the migratory abilities of the siHMGA2-transfected HCT116 cells. As shown in Fig. 2B, the migratory abilities significantly reduced in approximately 43% of siHMGA2-transfected HCT116 cells compared with those in the control cells. To investigate whether the phenotype of gene-specific or pharmaceutical Hsp90 inhibition is similar to that of gene-specific HMGA2 inhibition in the CRC cells, we performed cell proliferation and cell

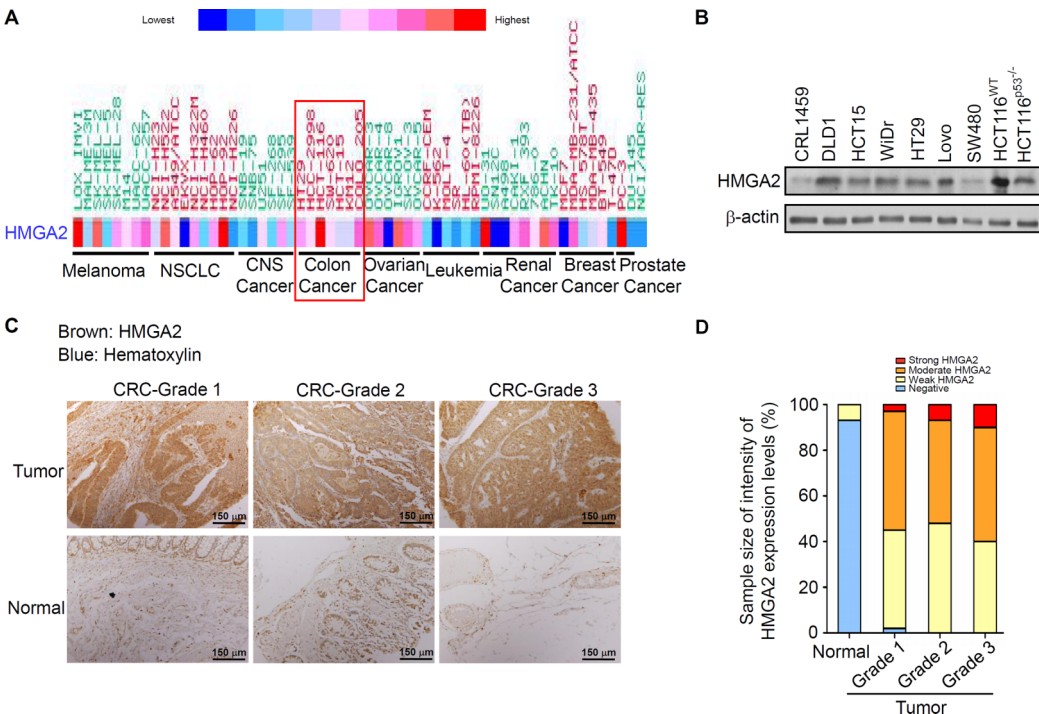

**Figure 1 HMGA2 was overexpressed in Colorectal Cancer (CRC) cell lines and tumors.** (A) Gene expression levels of HMGA2 protein in various human cancer cell lines. (B) HMGA2 protein analysis was conducted on proteins isolated from eight CRC cell lines and one noncancerous human colon cell line (CRL-1459). (C) Representative Immunohistochemical (IHC) images of HMGA2 expression on tissue microarray containing paired normal tissues and tumors of three CRC patients with different tumor grades. (D) HMGA2 protein expression levels obtained from the IHC results. The percentage of cases is plotted on the y-axis, and the type of sample is plotted on the x-axis; the color indicates the HMGA2 expression levels.

migration assays in HCT116 cells. siHsp90 knockdown significantly reduced mRNA expression level of Hsp90 about 50% (Fig. S4B). The proliferation index in siHsp90- and NVP-AUY922-treated HCT116 cells significantly reduced on Hsp90 knockdown compared with that in the control cells (Figs. 2C and 2E). Moreover, cell migration significantly reduced in the siHsp90- and NVP-AUY922-treated HCT116 cells. The inhibition rate was approximately 85% and 65% after siHsp90 and 40 nM NVP-AUY922 treatments, respectively (Figs. 2D and 2F, respectively). Thus, the effect of Hsp90 inhibition is similar to that of HMGA2 inhibition.

## Hsp90 regulates and interacts with HMGA2

Hsp90 is critical in regulating cell growth (*Cheung et al., 2010*; *Ko et al., 2012*; *Miyata, 2003*; *Nagaraju et al., 2014*), and HMGA2 has a well-documented role in this process (*Di Cello et al., 2008*; *Malek et al., 2008*; *Sun et al., 2013*; *Wend et al., 2013*); therefore, we examined whether the interaction between the Hsp90 and HMGA2 exists. Thus, we first performed RNA interference to deplete Hsp90 in the HCT116 cells and examined the effect of its depletion in the intracellular localization of both Hsp90 and HMGA2 by immunofluorescence analysis. As shown in Fig. 3A, Hsp90 siRNA-mediated endogenous

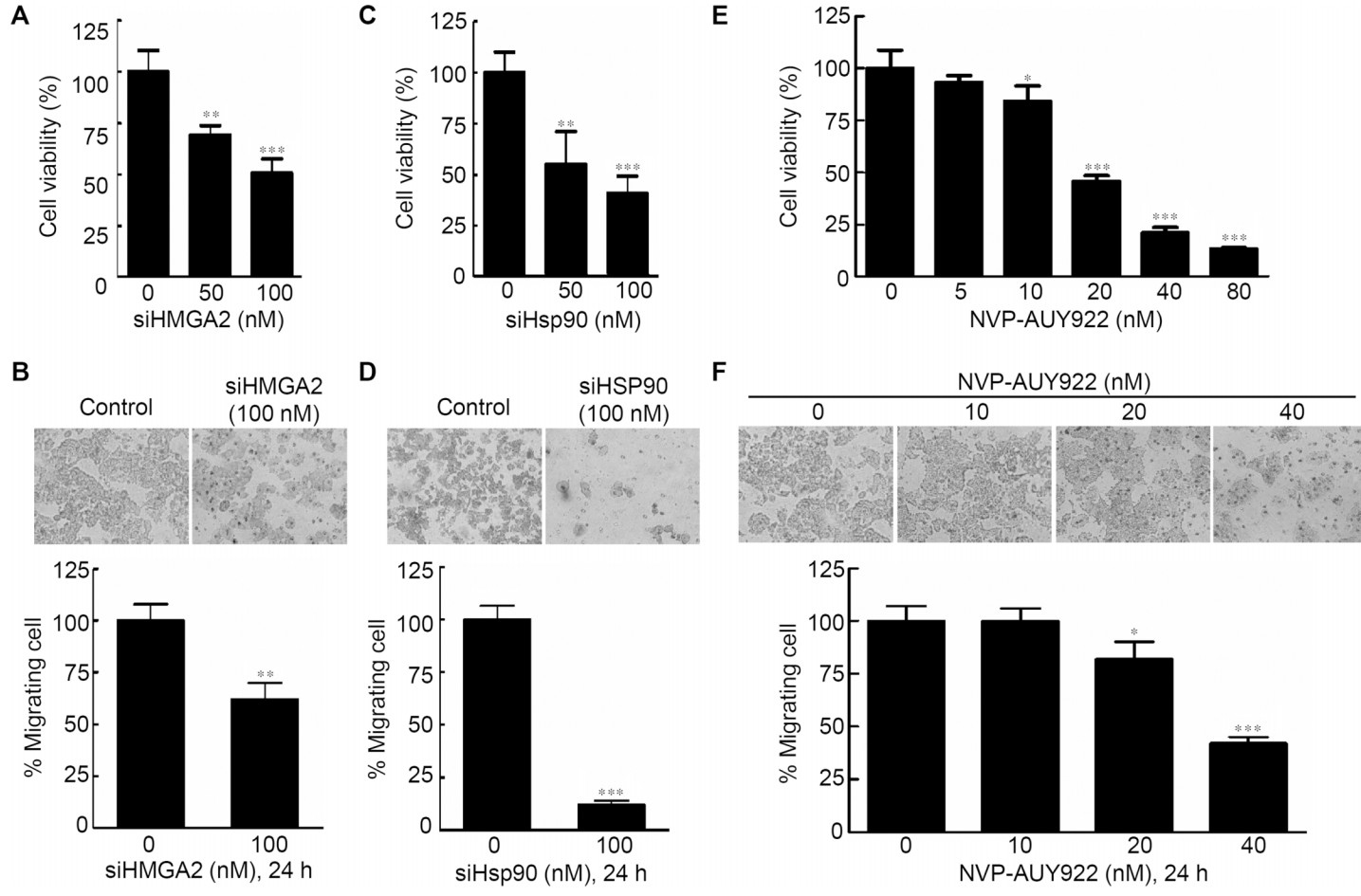

**Figure 2 Effects of gene-specific inhibition of HMGA2 or Hsp90 and pharmaceutical inhibition of Hsp90 were similar.** Cell viability assay (A, C, and E) and cell migration analysis (B, D, and F) were performed to determine the viability and migratory ability of HCT116 cells treated with siHMGA2, siHSp90, and various concentrations of NVP-AUY922 for 48 hours (cell viability assay) or 24 hours (cell migration assay), respectively. $*p < 0.05$, $**p < 0.01$, $***p < 0.001$. All experiments were performed in three independent experiments.

Hsp90 knockdown significantly reduced CDK4 (Hsp90 client protein) and HMGA2 expression, as well as induced Hsp70 (Hsp90 client protein) expression in the siHsp90-transfected HCT116 cells. Further, immunofluorescence result revealed that Hsp90 (red color) co-localized with HMGA2 (green color) in the nucleus of control siRNA-transfected HCT116 cells (Fig. 3B, control SiRNA row merged image show colocalization of Hsp90 and HMGA2; overlap of red and green: yellow). However, this phenomenon cannot observe in siHsp90 transfected HCT116 cells (Fig. 3B, siHsp90 (100 nM) row of merged image). These results indicated that the interaction between Hsp90 and HMGA2 exists. Next, we investigated whether the inhibition of HMGA2 through Hsp90-mediated inhibition by using Hsp90 inhibitor. As shown in Fig. 3C, HMGA2 protein expression was significantly reduced on NVP-AUY922-treated HCT116 cells at both 40 nM and 80 nM concentrations. To evaluate a potential Hsp90-HMGA2 interaction, we performed an immunoprecipitation assay to determine the effect of NVP-AUY922 on the physical interactions between Hsp90 and HMGA2. After NVP-AUY922 treatment, the HMGA2

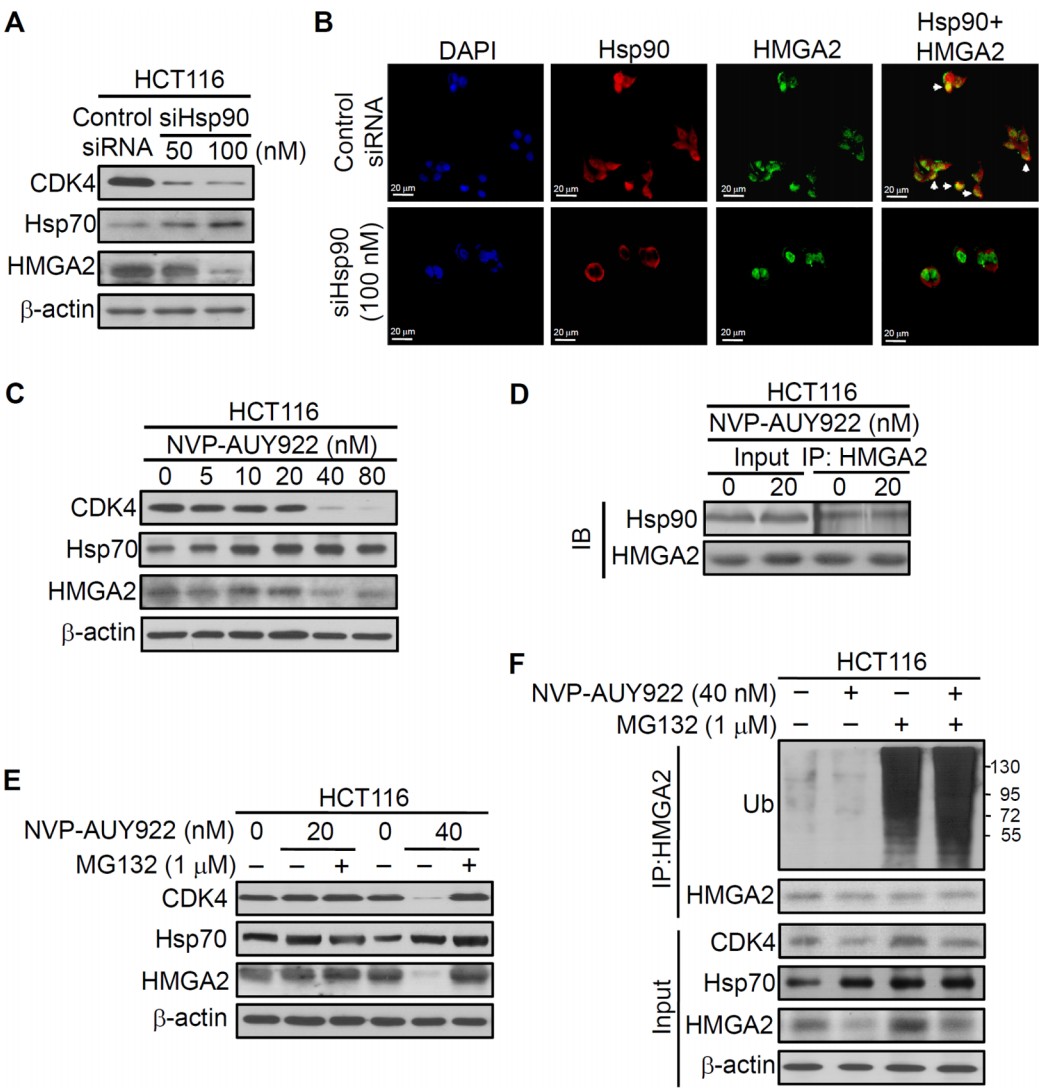

**Figure 3 Direct interaction between HMGA2 and Hsp90.** (A) CDK4, Hsp70, and HMGA2 were detected in siHsp90-transfected HCT116 cells. (B) Subcellular colocalization of HMGA2 and Hsp90. HCT116 cells were stained with DAPI (blue, nuclear stain) and antibodies to Hsp90 (red) or HMGA2 (green), and confocal images were acquired at 40× magnification. (C) HCT116 cells were treated with NVP-AUY922 at the indicated concentrations for 48 hours. Cell extracts were analyzed using Western blotting with the antibodies for CDK4, Hsp70, and HMGA2, respectively. (D) HCT116 cells were treated with NVP-AUY922 for 48 hours, HMGA2 was immunoprecipitated from 500-μg cell lysate, and resultant blots were probed for Hsp90 and HMGA2 antibodies, respectively. (E) The proteasome inhibitor MG132 (1 μM, 24 hours) protected against NVP-AUY922-facilitated suppression of HMGA2 expression in HCT116 cells. HCT116 cells were treated with 20 nM or 40 nM alone for 48 hours, or NVP-AUY922 pretreated for 24 hours and combination with MG132 for an additional 24 hours, and cell lysates were subjected to Western blot analysis using anti-CDK4, anti-Hsp70, anti-HMGA2, and anti-β-actin antibodies. (F) HCT116 cells were treated as above, and proteins extracts were Immuno-precipitated (IP) with anti-HMGA2. The ubiquitination of HMGA2 was analyzed by Western blotting with anti-ubiquitin. All experiments were performed in three independent experiments.

protein was immunoprecipitated with an anti-HMGA2 antibody and analyzed through Western blotting with anti-Hsp90 or anti-HMGA2 antibodies. As shown in Fig. 3D, a single band was detected using anti-Hsp90 antibody in immunoprecipitates or input
lysate from NVP-AUY922-treated HCT116 cells. In addition, the protein interaction between Hsp90 and HMGA2 was not affected by treatment with NVP-AUY922 at 20 nM. The interaction between Hsp90 and HMGA2 was also observed in DLD1 HMGA2-GFP cells to show that Hsp90 was coimmunoprecipitated by the GFP antibody in DLD1 HMGA2-GFP cells (Fig. S5). Hsp90 inhibitors cause degradation of Hsp90 client proteins via a proteasome-dependent pathway (*Basso et al., 2002*). Therefore, we examined whether proteasomal degradation mediates the loss of HMGA2 protein after treatment with NVP-AUY922. As shown in Fig. 3E, decreased levels of CDK4 and HMGA2 by NVP-AUY922 treatment at 40 nM were recovered by treatment with a proteasomal inhibitor, MG132, indicating the involvement of proteasomal degradation in this loss of HMGA2 protein. Previous study have demonstrated that Hsp90 inhibitor-mediated proteasomal degradation of Hsp90 client proteins was preceded by their ubiquitination (*Grbovic et al., 2006*); therefore, we then tested whether HMGA2 was ubiquitinated prior to its degradation in NVP-AUY922-treated cells. Immunoprecipitation of HMGA2 followed by Western blot analysis with an anti-ubiquitin antibody detected significantly higher levels of ubiquitinated HMGA2 in the presence of the combination of MG132 and NVP-AUY922, compared with either agent alone (Fig. 3F). Taken together, these data suggest that downregulation of HMGA2 protein was a direct effect of Hsp90 inhibition and also indicate that Hsp90 is necessary for the stability of HMGA2.

### Inhibition of HMGA2 protein increased sensitivity of Hsp90 inhibitor

HMGA2 contributes to resistance against anticancer drugs in various cancer cell lines (*Gyorffy et al., 2006*). Thus, HMGA2 silencing was hypothesized to increase the sensitivity to anticancer drugs in cancer cells. To test this hypothesis, the HCT116 cell line with elevated HMGA2 expression was selected for transfection with HMGA2 small-interfering RNA oligomer (siHMGA2) or scrambled oligomer (control siRNA). HMGA2 protein expression and cell viability were subsequently examined. As shown in Fig. 4A, HMGA2 protein expression was significantly inhibited in siHMGA2-transfected HCT116 cells. To examine the NVP-AUY922 drug sensitivity in siHMGA2-transfected HCT116 cells, a cell viability assay was performed. NVP-AUY922 treatment significantly reduced the cell viability of siHMGA2-transfected HCT116 cells compared with the control siRNA-transfected HCT116 cells (Fig. 4B). The HMGA2, CDK4, and Hsp70 proteins expression were examined in control siRNA or siHMGA2-transfected HCT116 cells in the presence of NVP-AUY922 (Fig. 4C). In contrast, we examined the NVP-AUY922 drug sensitivity in HMGA2-overexpressed CRC cells. A stable cell line, DLD1 HMGA2-GFP, was established and characterized using an anti-GFP antibody for Western blotting (Fig. 4D). As expected, the proliferation index of the stable DLD1 HMGA2-GFP cells significantly reduced on NVP-AUY922 treatment group compared with the parental group (Fig. 4E). The effect of NVP-AUY922 in inhibition of HMGA2 and CDK4 proteins expression was attenuated in HMGA2 stable clone (Fig. 4F). These results are consistent with the previous observation that the HMGA2 expression levels influence anticancer drug sensitivity.

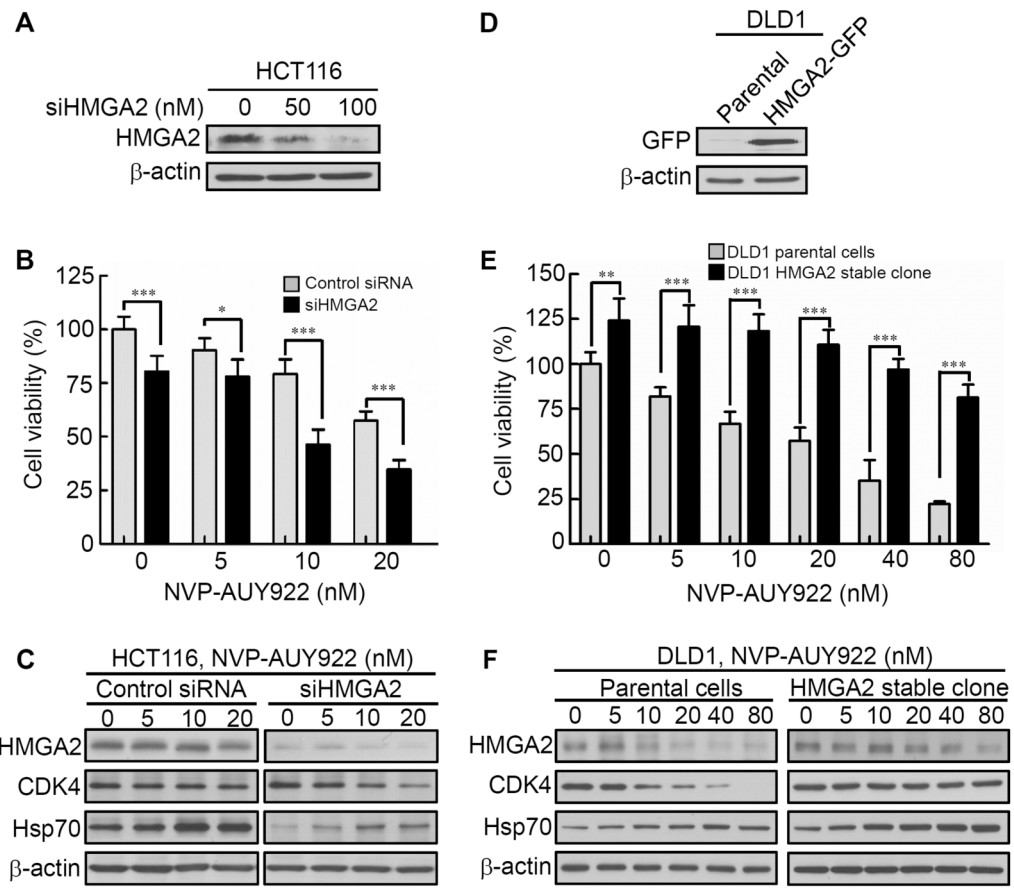

**Figure 4 Expression levels of HMGA2 are responsible for NVP-AUY922 drug sensitivity.**
(A) HMGA2 was detected in siHMGA2-transfected HCT116 cells. (B) HCT116 cells were transfected with control siRNA and siHMGA2 (100 nM) for 48 hours and subsequently incubated with NVP-AUY922 at the indicated concentrations for an additional 48 hours. A cell viability assay was performed to determine the viability of cells treated with various NVP-AUY922 concentrations. Bars, SD (n = 6). (C) Western blot analysis of proteins expression of HMGA2, CDK4, and Hsp70 in HCT116 cells transfected with control siRNA or siHMGA2 for 48 hours and subsequently incubated with NVP-AUY922 at the indicated concentrations for an additional 48 hours. (D) Western blotting with anti-GFP antibody of the parental and stable HMGA2-GFP groups of HCT116 cells. (E) Cell proliferation assays of the parental and stable HMGA2-GFP groups of HCT116 cells treated with NVP-AUY922 at the indicated concentrations for 48 hours. Bars, SD (n = 6). $^*p < 0.05$, $^{**}p < 0.01$, $^{***}p < 0.001$. (F) Western blot analysis of proteins expression of HMGA2, CDK4, and Hsp70 in NVP-AUY922-treated of parental or HMGA2-GFP stable expression of DLD1 cells. All experiments were performed in three independent experiments.

## HMGA2 as a master regulator of Epithelial–Mesenchymal Transition (EMT) and involved in NVP-AUY922-mediated suppression of EMT

To investigate the role of HMGA2 in regulating EMT of CRC cells, we investigated the effect of siRNA-mediated knockdown of HMGA2 on the expression of EMT effectors in HCT116 cells. As shown in Fig. 5A, HMGA2 knockdown inhibited EMT in HCT116 cells, which was evidenced by reduced HMGA2-regulated mesenchymal markers (Twist, Snail, and Slug) (*Li et al., 2014*; *Tan et al., 2012*; *Thuault et al., 2008*) as well as Vimentin

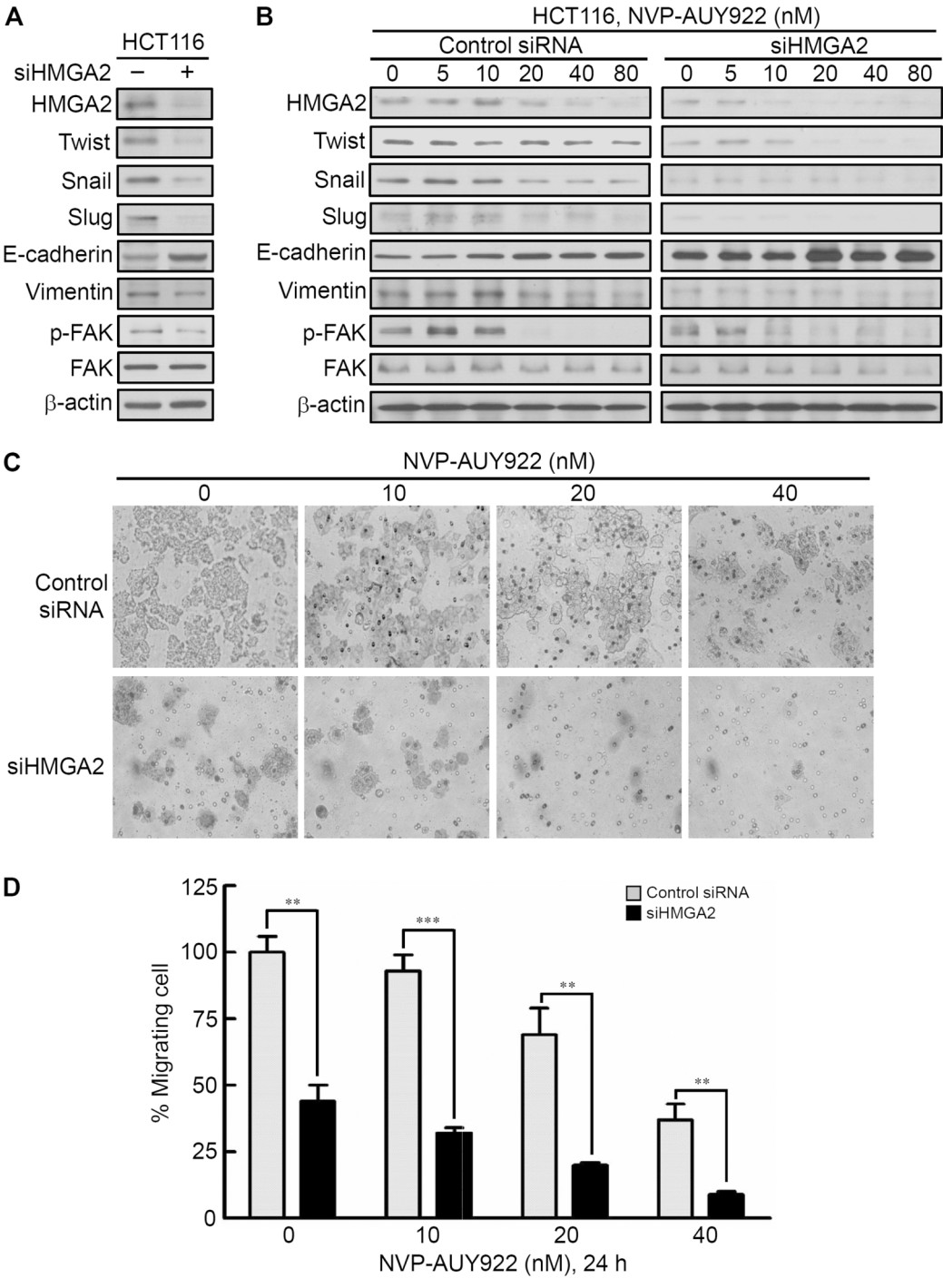

**Figure 5 Knockdown HMGA2 expression can enhance the effect of NVP-AUY922-mediated suppression of EMT and migratory ability of HCT116 cells.** (A) siRNA-mediated knockdown of HMGA2 inhibited HMGA2-regulated EMT in HCT116 cells, as revealed by loss of mesenchymal markers Twist, Snail, Slug, Vimentin, and reduction the phosphorylation level of FAK and gain of epithelial marker E-cadherin. (B) Effect of siRNA-mediated knockdown of HMGA2 on NVP-AUY922–mediated reversal of mesenchymal character in HCT116 cells. (C and D) Concentration-dependent effects of NVP-AUY922 on the migratory activity of HCT116 cells after 24 hours of treatment. $^{**}p < 0.01$, $^{***}p < 0.001$. All experiments were performed in three independent experiments.

expression in conjunction with concomitant increases in the expression of the E-cadherin. In addition, Focal Adhesion Kinase (FAK) activation is important for cancer motility. It has demonstrated that FAK is regulated by HMGA2 in melanoma cells (*Zhang et al., 2015a*). siHMGA2 treatment attenuated the phosphorylation of FAK without affecting the total FAK in HCT116 cells (Fig. 5A). Pursuant to these findings, we used the siRNA-mediated knockdown of HMGA2 to verify its effect in the NVP-AUY922-mediated suppression of EMT in HCT116 cells. As shown in Fig. 5B, knockdown HMGA2 expression can enhance the effect of NVP-AUY922-mediated suppression of EMT in HCT116 cells to compare with control siRNA group. Next, the in vitro efficacy of NVP-AUY922 in suppressing cancer cell mobility was illustrated by its dose-dependent inhibition of the migration of HCT116 cells-transfected with control siRNA or siHMGA2 after 24 hours of treatment in transwell assays. As shown in Figs. 5C and 5D, the migratory abilities significantly reduced about 50% of siHMGA2-transfected HCT116 cells compared with those in the control cells and the number of migrating cells was significantly reduced in siHMGA2-transfected HCT116 cells-treated with NVP-AUY922. Together, these findings suggest that NVP-AUY922 can enhance the reduction of EMT in siHMGA2-transfected HCT116 cells.

### Phospho-kinase array for investigating NVP-AUY922-induced altered activity of kinases that regulate growth and mobility of HCT116 cells

Several Hsp90 inhibitors have been identified to target Hsp90 client proteins, such as receptors, kinases, and transcription factors, which are involved in oncogenesis (*Porter, Fritz & Depew, 2010*; *Trepel et al., 2010*). Extracellular Signal-Regulated Kinase (ERK) and FAK have been demonstrated to be regulated by Hsp90 and were involved in HMGA2-regulated CRC cell growth and mobility (*Chen et al., 2010*; *Li et al., 2013*; *Li et al., 2014*; *Ory et al., 2015*; *Zhang et al., 2015a*). Our aforementioned results (Fig. 3) demonstrated that Hsp90 might be the upstream regulator of HMGA2. Therefore, using the human phospho-kinase array, we examined whether NVP-AUY922 treatment in HCT116 cells altered the activity of kinases involved in regulating HMGA2. As shown in Fig. 6A, the phosphorylation levels of ERK, FAK, and CREB were significantly inhibited in NVP-AUY922-treated HCT116 cells. The CREB/HMGA2 pathway is crucial in malignant transformation (*Shibanuma et al., 2012*). Furthermore, CREB is a transcription factor and a downstream target of the ERK pathway (*Qi et al., 2008*). Accordingly, we hypothesized that HMGA2-regulated cell growth can be inhibited using NVP-AUY922 treatment through NVP-AUY922-regulated ERK-CREB-HMGA2 signaling. Thus, HCT116 cells were dose-dependently treated with NVP-AUY922 for 48 hours, and the phosphorylation status and total protein expression levels of ERK and CREB were examined. As shown in Fig. 6B, Western blotting results revealed that the phosphorylation status and total protein expression levels of ERK and CREB were significantly inhibited in the NVP-AUY922-treated HCT116 cells. To further investigate whether ERK was indeed involved in regulation of HMGA2, we examined the dose effect of AZD6244, a potent ERK inhibitor, on the phosphorylation status of ERK, CREB and HMGA2 proteins expression of HCT116 cells. As shown in Fig. 6C, Western blotting indicated that

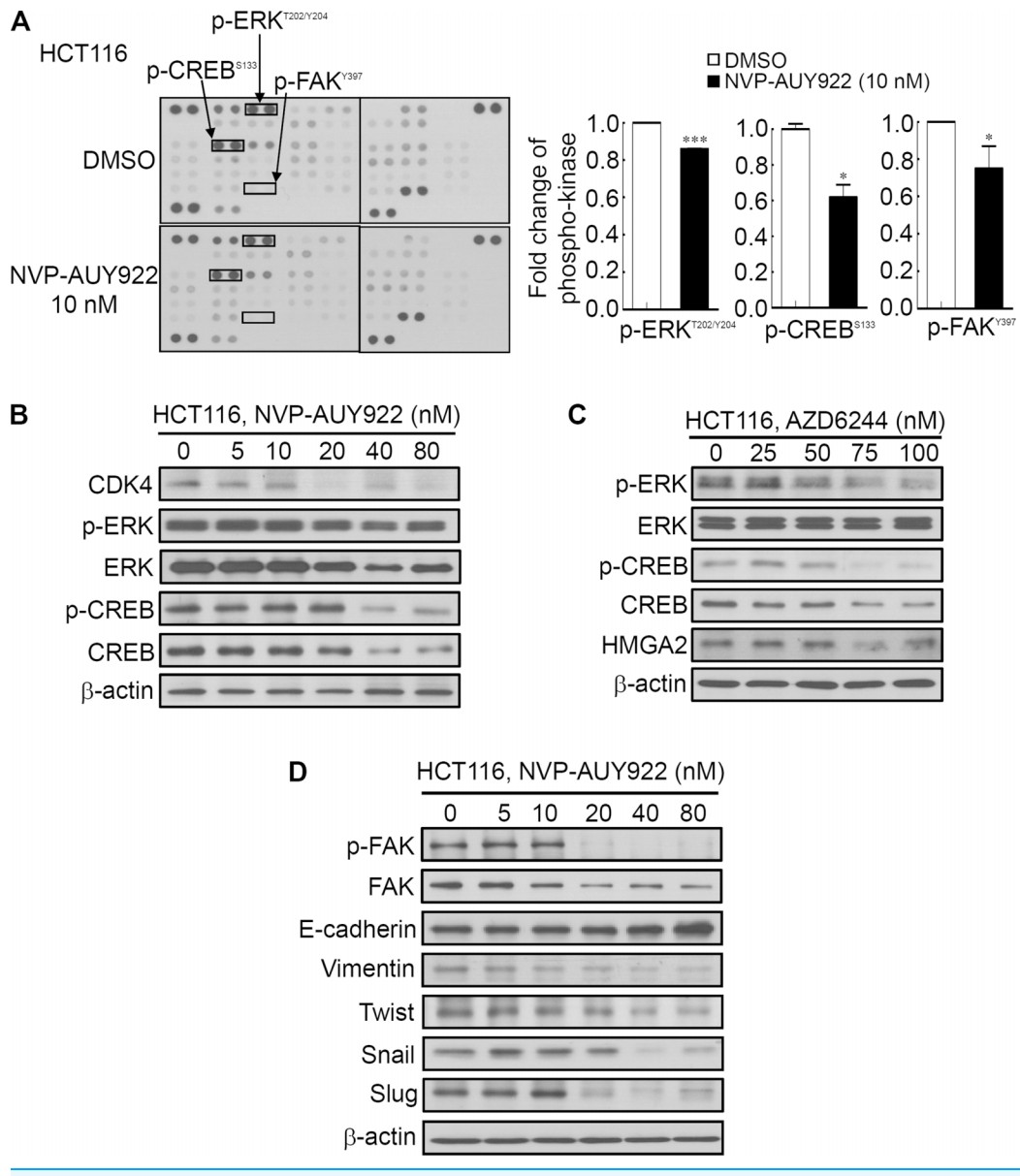

**Figure 6 Human phospho-kinase array analysis in response to NVP-AUY922 treatment in HCT116 cells.** (A) HMGA2-associated kinases, ERK, CREB, and FAK, were significantly downregulated on NVP-AUY922 treatment. (B) Western blotting results of the concentration-dependent effects of NVP-AUY922 on the phosphorylation and expression of ERK and CREB in HCT116 cells. (C) Western blotting revealed the dose effect of AZD6244 on the phosphorylation status of ERK and CREB as well as HMGA2 proteins expression of HCT116 cells. (D) Western blotting results of the concentration-dependent effects of NVP-AUY922 on the phosphorylation and expression of FAK and various EMT effectors of HCT116 cells. All experiments were performed in three independent experiments.

phosphorylation status of ERK and CREB was significantly inhibited in AZD6244-treated HCT116 cells at both 75 nM and 100 nM in conjunction with concomitant the decrease expression of HMGA2. Downregulated FAK expression results in the loss of mesenchymal markers and increased expression of the epithelial marker, E-cadherin, in breast tumor models (*Kong et al., 2012*). In addition, Hsp90 inhibition disrupts FAK signaling and

inhibits tumor progression (*Schwock et al., 2009*). To understand whether FAK was involved in Hsp90-regulated EMT signaling, the phosphorylation status of FAK and EMT effectors were examined. NVP-AUY922 dose-dependently reduced the phosphorylation level of FAK, accompanied by parallel changes in the expression of various EMT effectors, including E-cadherin, Vimentin, Twist, Snail, and Slug in HCT116 cells, with the reversal from a mesenchymal to an epithelial phenotype (Fig. 6D). This result is consistent with the findings of Fig. 5A to indicate Hsp90-regulated EMT signaling through HMGA2-regulated signaling. Collectively, these results clearly indicate that Hsp90 can indirectly regulated HMGA2 via activation of the ERK signaling pathway, and this regulatory mechanism can be inhibited by treatment the Hsp90 inhibitor.

## DISCUSSION

HMGA2 overexpression in various human neoplasias is associated with highly malignant phenotypes, such as chemoresistance, metastasis, and poor survival (*Di Cello et al., 2008*; *Mahajan et al., 2010*; *Wang et al., 2011*; *Yang et al., 2011*). HMGA2 or HMGA2-regulated signaling is the preferred therapeutic target in CRC. This is the first study to recognize HMGA2 as a newly identified Hsp90 client protein and to propose pharmacological Hsp90 inhibition as a promising strategy for impairing HMGA2 function. We demonstrated that the Hsp90 mRNA expression levels in primary and metastatic CRC tissues were similar to those of HMGA2, analyzed from the Gene Expression Omnibus repository (GSE21815), and reported that the Hsp90 inhibitor follows a rational therapeutic approach in inhibiting HMGA2-triggered tumorigenesis. The knockdown of Hsp90 using Hsp90 siRNA significantly reduced HMGA2 expression, and the effects of Hsp90 and HMGA2 knockdown were similar. The relationship of HMGA2 and Hsp90 was examined by immunofluorescence and in vitro ubiquitination assays in CRC cells. Moreover, our cell viability data clearly demonstrated that HMGA2 expression levels influenced NVP-AUY922-induced drug sensitivity of the CRC cells. NVP-AUY922 treatment in CRC cells significantly downregulated the regulatory activities of kinases involved in regulation of HMGA2. Collectively, this is the first study to report that Hsp90 inhibitor significantly suppressed HMGA2 protein expression and HMGA2-mediated regulation of cell growth and mobility.

MiRNAs are critical in the regulation of HMGA2 protein expression (*D'Angelo, Esposito & Fusco, 2015*). Let-7a is one of the most critical tumor suppressor miRNA that regulates HMGA2 expression (*Wang et al., 2013*; *Wu et al., 2015*; *Yang et al., 2015*). In particular, let-7a dysregulation was observed in CRC (*Pallante et al., 2015*). In the present study, let-7a expression was significantly induced using NVP-AUY922 (40 nM) treatment in HCT116 cells (Fig. S6A), and HMGA2 protein expression was simultaneously inhibited (comparison of Figs. 3C and S6A). It has shown that the biogenesis of let-7a was blocked by overexpression of c-Myc/Lin28B axis in cancer cells (*Pang et al., 2014*). In addition, it has been demonstrated that Stat3-coordinated Lin28B–let-7–HMGA2 signaling to circuit initiate and maintain oncostatin M-driven EMT (*Guo et al., 2013*). To determine whether reactivation of let-7a by treatment with Hsp90 inhibitor through inhibition of c-Myc/Lin28B axis or Stat3 signaling, these

proteins were detected in NVP-AUY922-treated HCT116 cells. As shown in Fig. S6B, the phosphorylation status of Stat3 and protein expression of Lin28B and c-myc were completely inhibited on NVP-AUY922-treated HCT116 cells at 40 nM for 24 hours. In clinical CRC specimens, quantitative RT-PCR and IHC analysis revealed downregulated let-7a expression levels and upregulated HMGA2 protein expression levels, respectively (data not shown). These results show that let-7a acts as a suppressor of CRC tumorigenesis, and NVP-AUY92-induced let-7a reactivation can inhibit HMGA2-triggered cell growth and mobility of CRC cells.

HMGA2 is an architectural transcription factor and belongs to the high motility group A family. This family of proteins can modify the structure of its binding partners to generate a conformation that facilitates various DNA-dependent activities and influences various biological processes, including cell growth, metastasis, and survival (*Califano et al., 2014*; *Morishita et al., 2013*). HMGA2 protein regulates the transcription of several EMT-related genes and thus is closely associated with tumor invasion and metastasis (*Morishita et al., 2013*). HMGA2 upregulated the expression of Snail and Twist and downregulated the expression of E-cadherin in normal murine mammary gland epithelial cells (*Thuault et al., 2006*). In addition, HMGA2 positively regulated Slug expression by directly binding to the regulatory region of the Slug promoter (*Li et al., 2014*). HMGA2 was involved in cordycepin-mediated suppression of late-stage melanoma metastasis through the modulation of the activation of FAK and expression of EMT effectors (*Zhang et al., 2015b*). Furthermore, FAK expression downregulation results in the loss of mesenchymal markers and increased epithelial marker expression in breast tumor models (*Kong et al., 2012*). These results reveal the criticality of HMGA2 in cancer progression, and thus HMGA2 is a potential molecular target for preventing cancer progression. However, the molecular mechanism of the Hsp90 inhibitor in the inhibition of metastasis remains unclear. An Hsp90 inhibitor, 17-allylamino-17-demethoxygeldanamycin, inhibited prostate cancer metastasis through Slug inhibition (*Ding et al., 2013*). This is the first study to examine the potency of a second generation Hsp90 inhibitor, NVP-AUY922, on the inhibition of migration in CRC cells through the simultaneous inhibition of EMT effectors regulated by HMGA2.

In summary, NVP-AUY922 reduced the activity and expression of ERK and CREB and suppressed CRC cell growth. In addition, NVP-AUY922 downregulated the expression of HMGA2 and HMGA2-mediated EMT effectors, which suppressed cell motility, suggesting that NVP-AUY922 not only regulates the growth of CRC cells but also its dissemination.

## CONCLUSIONS

Our study is the first to identify the interaction between Hsp90 and HMGA2 and that the Hsp90 inhibitor has therapeutic potential to inhibit HMGA2-triggered tumorigenesis. Moreover, our findings clarify the downregulation of HMGA2 was a direct effect of Hsp90 inhibition and also indicate that Hsp90 is necessary for the stability of HMGA2. Moreover, Hsp90 inhibitor also can indirectly regulated HMGA2 via inactivation of the ERK signaling pathway or reactivation of let-7a.

### Funding

This project was funded by the Ministry of Science and Technology of Taiwan (MOST 104-2320-B-038-006 to KHL), Taipei Medical University (TMU101-AE1-B19 and TMU104-AE2-I02-5 to KHL), Taipei Medical University-Shuang Ho Hospital (104TMU-SHH-10 to KHL) and the proposal of the Ministry of Health and Welfare (MOHW 104-TDU-B-212-124-001 to YY). The funders had no role in study design, data collection and analysis, decision to publish, or preparation of the manuscript.

### Grant Disclosures

The following grant information was disclosed by the authors:
Ministry of Science and Technology of Taiwan: 104-2320-B-038-006.
Taipei Medical University: TMU101-AE1-B19 and TMU104-AE2-I02-5.
Taipei Medical University-Shuang Ho Hospital: 104TMU-SHH-10.
Ministry of Health and Welfare: 104-TDU-B-212-124-001.

### Competing Interests

The authors declare that they have no competing interests.

### Author Contributions

- Chun-Yu Kao conceived and designed the experiments, performed the experiments, analyzed the data, contributed reagents/materials/analysis tools, wrote the paper, prepared figures and/or tables.
- Pei-Ming Yang performed the experiments, analyzed the data, contributed reagents/materials/analysis tools, prepared figures and/or tables.
- Ming-Heng Wu performed the experiments, analyzed the data, contributed reagents/materials/analysis tools, prepared figures and/or tables.
- Chi-Chen Huang performed the experiments, analyzed the data, contributed reagents/materials/analysis tools, prepared figures and/or tables.
- Yi-Chao Lee conceived and designed the experiments, analyzed the data, contributed reagents/materials/analysis tools, wrote the paper, prepared figures and/or tables, reviewed drafts of the paper.
- Kuen-Haur Lee conceived and designed the experiments, analyzed the data, contributed reagents/materials/analysis tools, wrote the paper, reviewed drafts of the paper.

### Data Deposition

The research in this article did not generate any raw data.

### Supplemental Information

Supplemental information for this article can be found online at http://dx.doi.org/10.7717/peerj.1683#supplemental-information.

# PeerJ

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
