# Peer review of "Heat shock protein 90 is involved in the regulation of HMGA2-driven growth and epithelial-to-mesenchymal transition of colorectal cancer cells"

_PeerJ, doi:10.7717/peerj.1683_

## Round 0.1 · original submission · Major Revisions

After careful consideration, I believe that your study has the potential to be published provided you revise several fundamental aspects of your paper, as listed by reviewer #2 and #3. In particular you should concentrate on the experimental design and provide adequate control and complementary assays. If you are prepared to undertake the work required, I would be pleased to reconsider my decision.

·

Basic reporting

No Comments

Experimental design

No Comments"

Validity of the findings

No Comments"

Additional comments

In this manuscript entitled “Heat shock protein 90 is involved in the regulation of HMGA2-driven growth and epithelial-to-mesenchymal transition of colorectal cancer cells” the authors describe the interaction between HSP90 and HMGA2 proteins. Moreover the authors describe the use of HSP90 inhibitors to suppress HMGA2 expression and HMGA2-mediated EMT effectors, which suppressed cell motility. The study is of interest and well described. Only for completeness of the paper, we suggest some revisions:
- The authors should further describe and discuss the IP experiment presented in Figure 3C;
- The authors should explain why the concentration of NVP-AUY922 used in Figure 3C is 0-20 nM, while in Figure 3B the effects of the inhibitor are shown at higher concentrations.

Reviewer 2 ·

Basic reporting

The resolution and magnification of Fig. 1C and Fig. S2 is not fit for publication. More clear images should be supplied by the authors and the pathological structure should be recognized in the images.

Experimental design

1. As shown in Fig. 4, although the effects of gene-specific inhibition of HMGA2 or Hsp90 and pharmaceutical inhibition of Hsp90 were similar, we still cannot get the conclusion that there is some relationship between HMGA2 and Hsp90. More solid evidence is required.
2. As shown in Fig. 5, the expression and phosphorylation of some kinases were detected in response to different concentration of NVP-AUY922 treatment. Although some kinases were down-regulated after the treatment of NVP-AUY922 in high concentration, we still cannot get the conclusion that there is some kind of relationship among these kinases. The evidence is not enough and more experimental groups is required.
3. As shown in Fig. S1, the relationship of HMGA2 or Hsp90 mRNA expression with clinical stages was analyzed separately and it was significant in public database, however, it was more important to analyze the relationship of HMGA2 and Hsp90.

Validity of the findings

1. According to the hypothesis of the authors, the Hsp90 inhibitor, NVP-AUY922 was able to down-regulate the expression of HMGA2 through a let-7a dependent way, and subsequently influenced downstream signal pathway. As shown in Fig. 3D, the reactivation effect of NVP-AUY922 on let-7a was significant only when its concentration was more than 40 nm. However, as shown in Fig 2A, the effect of NVP-AUY922 on cell viability was also significant in low concentration. These results did not support the authors’ hypothesis.
2. The IHC images should be evaluated by professional pathologist. As shown in Fig. S2, it only presented the proportion of carcinoma nest, not the expression level of HMGA2.

Additional comments

The manuscript entitled “Heat shock protein 90 is involved in the regulation of HMGA2-driven growth and epithelial-to-mesenchymal transition of colorectal cancer cells” presents some results about the effect of Hsp90 inhibition on HMGA2 protein expression and subsequent signal pathway. The authors want to demonstrate that the treatment of NVP-AUY922, an Hsp90 inhibitor, was able to regulate the expression of HMGA2, and influence cell growth and migration of CRC cells in a HMGA2 dependent pathway. The effect of NVP-AUY922 on HMGA2 expression occurred through the reactivation of let-7a. However, the results were not enough to support the authors’ hypothesis and there are a number of issues with these experiments that need to be addressed.

Reviewer 3 ·

Basic reporting

The manuscript by Kao et al. describes a new role for Hsp90 in regulating HMGA2 expression and Hsp90 inhibitor NVP-AUY922 could be used to downregulate HMGA2 to inhibit the proliferation and migration of colorectal cancer cells. In addition, the authors have also shown that NVP-AUY922 inhibits EMT by upregulating E-cadherin expression and downregulation of mesenchymal markers in CRC cells.

In terms of format, it would be ideal to revise some parts:
- There are some almost identical sentences used in both abstract and introduction sections.
- The supplementary data should be moved from the introduction section to the results section.
- There are paragraphs in the manuscript that are ambiguous and not clearly written.

The method section is inadequate, and therefore, additional details should be added. It is also unclear how many times and replicates for each experiment had been performed.
1. Include the names of institution affiliated with the scientists providing the cell lines.
2. List the type of antibiotics used in the cell culture media.
3. Indicate if TaqMan probes or SybrGreen is used. Primer sequences are missing.
4. Immunoprecipitation protocol is missing.
5. Transfection protocol is missing; how long was the transfection? List oligo sequences for siHsp90 and siHmga2.
6. It is written ‘standard error’ in the method section but elsewhere in the manuscript, values are often reflected as standard deviation. Which one of them is used?

Experimental design

See below.

Validity of the findings

The authors have shown that Hsp90 has a new role in regulating HMGA2 expression. However, the data remains incomplete due to lack of adequate controls and complementary assays. There was no concrete evidence indicating HMGA2’s main role in driving EMT in HCT116 cells, bearing in mind that NVP-AUY922 have multiple effects on other oncogenic proteins, of which does not represent the title of the manuscript. Therefore, it is still early to draw conclusions from the data.

Major points:
- To strengthen the notion that HMGA2 confers resistance in CRC, include another cell line that is more resistant to NVP-AUY922, e.g. HT29 (based on their previous work); examine HMGA2 levels and then knockdown HMGA2 to see if the cell line becomes sensitized towards NVP-AUY922. HMGA2 WBs should be included for both HCT116 and DLD1-GFP-HMGA2 cells treated with/without NVP-AUY922 (Fig. 2).
- CDK4 appears to be a poor readout for the efficacy of NVP-AUY922 treatments in this study. Authors should consider to include additional surrogate marker of Hsp90 inhibition, e.g. Hsp70 (Dakappogari, et al., Biomarker, 2010). Fig. 3A & 3B: Hsp90 WB to show knockdown efficiency and also protein levels after treatment with NVP-AUY922. How long was the siRNA transfection? CDK4 WB quality is poor and can be improved.
- The interaction between Hsp90 and HMGA2 is not convincing, as there are no IgG controls or WB for HMGA2. In the text (page 8, line 8), how is HMGA2 expression observed if HMGA2 WB is missing? Assuming that the IP experiment worked, authors did not discuss the discrepancies of Hsp90-HMGA2 interaction observed between the two cell lines used, HCT116 and HCT15. Can this interaction be also observed in their overexpression system, i.e. DLD1-GFP-HMGA2 cells?
- Hsp90 activity is required for many kinases, e.g. ERK, FAK, and Hsp90 negatively regulates let-7a expression (in this study). Therefore, pharmacological inhibition of Hsp90 could be an indirect effect on HMGA2 regulation. Is HMGA2 a bona fide client protein of Hsp90? This is not shown or discussed. The interaction study between Hsp90 and HMGA2 is not sufficient to prove this.
- Fig. 3D: NVP-AUY922 treatments were for 24 h, in contrast to 48 h in other experiments, and the increment of let-7a expression was small, 1.5 fold at 40 nM. Is there a reason for the shorter treatment and how many times have the experiment been done? Since let-7a affects HMGA2 at the post-transcriptional level, what are the protein levels of HMGA2 at the 24-h time point?
- Fig. 5 comes back to my earlier point on Hsp90 being upstream of many kinases, which are also known to activate the EMT program, independent of HMGA2. Supplementary data should be included to show that HMGA2 is the main driver of EMT in HCT116 cells. Authors attempt to put forth a Hsp90-ERK-CREB-HMGA2 pathway in CRC cells; can they explain the high levels of p-ERK and p-CREB at 10 nM (even at 20 - 80 nM) NVP-AUY922, when HMGA2 is already downregulated? p-FAK and EMT effectors (Twist, Snail and Slug) levels are also high at 10 nM NVP-AUY22. Therefore, the data is not convincing to conclude that NVP-AUY922 could inhibit HMGA2-mediated EMT or HMGA2 regulation by its ‘associated kinases’ in HCT116 cells. Since HMGA2 confer resistance to NVP-AUY922 in CRC, have the authors check the combinatorial effects of NVP-AUY922 and siHmga2 treatment on cell proliferation, migration and EMT?

Minor points:
- Fig 1B, 2A: mRNA levels should be validated by qPCR to complement protein expression.
- Fig. 3 did not show any assays that measure HMGA2 activity, please re-phrase the subheading in the figure legend.
- Fig. 4A & 4C: How long was the siRNA transfection when the cell viability assay was done and show qPCR or WB to indicate knockdown efficiency.
- Fig. 4E, 5B & 5C: How long was NVP-AUY922 treatment?
- Fig. 5A: This is an array which measures the phosphorylation status of kinases and not correctly described in the figure legend. Indicate what was used for normalisation for the bar graphs.
- Page 6, line 24: should be 9 different types of cancers.
- Page 8, line 22: “epithelial-state transition” should be “epithelial-mesenchymal transition”.
- Authors should be cautious in their discussion regarding the use of NVP-AUY922 in inhibiting metastasis of CRC cells because they have not done any in vivo experiments, nor referred to other literature (page 11, line 17).

Additional comments

The study is interesting, relevant to the field and worth further investigation. Authors could focus on the mechanistic aspect of HMGA2 as a client protein of Hsp90, which is the novelty of this paper. It would be also interesting to know how does Hsp90 regulate let-7a.
However, the current finding is still preliminary and requires a major revision before it can be considered for publication. The manuscript can be further improved in terms of clarity and flow.

---

## Round 0.2 · Minor Revisions

This improved version of the manuscript will likely be suitable for publication if it is revised to address the point of reviewer 3 listed below.

Reviewer 3 ·

Basic reporting

The manuscript is an improved version and adheres to PeerJ criteria. The authors have addressed most of the reviewers’ concerns satisfactorily, to which I would recommend acceptance. I have some minor suggestions for the authors, please see below.

Experimental design

no comments.

Validity of the findings

The authors have shown that HMGA2 is regulated by ERK signaling (Fig. 6C). There could be alternative pathways, since NVP-AUY922 treatment has very subtle effect on the phospho-ERK levels in HCT116 cells (Fig. 6B), which implies that Hsp90 regulation of HMGA2 could be independent of ERK. In other words, Fig.6B is not in total agreement with the authors’ conclusion (lines 375–377, 444).

Additional comments

Minor points

line 179: protein ‘concentration’ instead of ‘content’

line 271: The following sentence, “…… the interaction between Hsp90 and HMGA2 exists.”, is suggested to make it clear for readers.

line 274: “… cells at both 40 nM and 80 nM concentrations.”

line 310: “… in the presence of NVP-AUY922…”

line 364: “… inhibited in AZD6244-treated HCT116 cells at both 75 nM and 100 nM …”

line 680: ’s’ is missing in ‘significantly’.

line 648: “… with 20 nM or 40 nM NVP-AUY922 alone for …”

Figures 2B, 2D and 2F: It is reflected “24 h” on the x-axis labels of the bar graphs, while figure legend indicated 48-h.

Figure 4A: please indicate in the figure legend (or key legend on the bar graph) if 50 nM or 100 nM siRNA was used in the cell viability assay.

---

## Round 0.3 · accepted · Accept

Dear authors, this revised version of the manuscript is now suitable for publication.